# Impact of Sizes of Servings, Glasses and Bottles on Alcohol Consumption: A Narrative Review

**DOI:** 10.3390/nu14204244

**Published:** 2022-10-12

**Authors:** Eleni Mantzari, Theresa M Marteau

**Affiliations:** Behaviour and Health Research Unit, University of Cambridge, Cambridge Public Health, East Forvie Site, Cambridge CB2 0SR, UK

**Keywords:** alcohol, wine, beer, consumption, serving size, portion size, glass size, bottle size

## Abstract

This review summarises the evidence on the impact of serving and container size on how much people drink, interventions that have the potential to reduce alcohol consumption across populations, thereby improving health. A rapid search identified 10 published reports of 15 studies and 1 review. Four studies focused on serving size, eight studies and the review on glass size, two studies on bottle size and one on both glass and bottle size. Twelve studies and the review focused on wine, one study on beer and two on both. All were conducted in England, by just two research groups. Removing the largest serving size of wine decreased wine sales by 7.6% (95% CI −12.3%, −2.9%) in a study in 21 licenced premises, reflecting findings from two prior studies in semi-naturalistic settings. Adding a serving size for beer that was a size smaller than the largest was assessed in one study in 13 licenced premises, with no evident effect. Reducing the size of wine glasses in restaurants decreased wine sales by 7.3% (95% CI −13.5%, −1.5%) in a mega-analysis of eight datasets from studies in five licensed premises. Using smaller wine glasses at home may also reduce consumption, but the evidence from just one study is less certain. No studies have assessed the impact of glass size for drinking beer. The effect of bottles smaller than the standard 750 mL on wine consumed at home was assessed in two studies: 500 mL bottles reduced consumption by 4.5% (95% CI −7.9%, −1.0%) in one study, but in another, using 375 mL bottles there was no evident effect. No studies assessed the impact of bottle or other container size for drinking beer. Reducing the size of servings, glasses and bottles could reduce wine consumption across populations. The impact of similar interventions for reducing consumption of other alcoholic drinks awaits evaluation. Further studies are also warranted to assess the generalisability of existing evidence.

## 1. Introduction

Excess alcohol consumption is a major contributor to premature death and disease globally. It has been linked to about 3 million annual deaths globally and is responsible for an estimated 5.1% of the global burden of disease [1]. Although reducing alcohol consumption is a global priority for decreasing the incidence of non-communicable diseases [2], as well as obesity [3,4,5,6], many countries have largely failed to meet relevant targets [7]. Reflecting this, in September 2022, WHO Europe made a historic decision to commit all 51 member states to a comprehensive plan to accelerate action to reduce alcohol consumption as a public health priority across Europe [7,8].

Interventions that target aspects of the physical environments that cue unhealthier behaviours have potential for scalable impacts at a population level, including on reducing excess alcohol consumption [9,10]. To date, most of the focus has been on policies that involve increasing the price of alcoholic drinks, and controlling their marketing and licensing [11,12]. An additional set of interventions, which have the potential to add to the effectiveness of these, involves changing the size of portions and containers of products that can harm health [9], including alcohol [1].

The size of portions, packages, and tableware has increased in recent decades [13,14]. A Cochrane systematic review conducted in 2015 found that people consume more food and non-alcoholic drinks when presented with larger portions or packages, and when using larger items of tableware, such as plates or glasses [9]. 

This well-documented “portion size effect” for food has, until recently, been neglected as a focus of study for alcohol consumption. There are no a priori reasons, however, why reducing the size of the servings and containers for alcohol—drinking vessels (e.g., glasses) and pouring vessels (e.g., bottles)—will not have a similar effect in reducing consumption. We summarise for the first time the evidence focusing on this potential. In doing so, we consider two dimensions of size relevant to the consumption of alcohol: first, the serving size of an alcoholic drink, and second, the container size of an alcoholic drink, encompassing both drinking and pouring vessels.

## 2. Materials and Methods

We conducted a rapid review using Google Scholar and platforms for the pre-registration of studies (ISRCTN and Open Science Framework (OSF)) to identify studies assessing the impact of the size of servings and containers on alcohol consumption. The search terms used were “alcohol”, “serving”, “glass”, “bottle” “carafe” “can”, “box” or “keg” in combination with “size”. One author (EM) screened through the ten first pages of results for each search. We also conducted backward and forward citations of the eligible studies retrieved and contacted authors of salient studies to enquire about any further potentially eligible research. 

The criteria for eligible studies, using the PICO framework [15], were: 

*Population:* adults.

*Intervention:* reduced (or increased) a serving size of a single drink of alcohol (i.e., not a size intended for sharing), the size of a glass for drinking alcohol, or a container holding alcohol, e.g., bottle, carafe, can, box).

*Comparison:* a different sized serving, glass or container to the intervention.

*Outcome*: observed (i.e., not self-report) measure of alcohol consumption. These comprised direct measures such as observation from video footage, and indirect measures such as sales.

Eligible studies included primary research and reviews (e.g., meta-analyses or mega-analyses). Only findings available in the public domain (published in peer reviewed journals, available as preprints or in the grey literature) were included. Studies identified through pre-registration platforms were included when the full data were available. We excluded studies that focused on changes to the shape of containers.

The titles, abstracts and full texts of potentially eligible reports were screened by one author (EM) with the second author (TM) involved in the inclusion/exclusion decision. Data were extracted by one author (EM).

## 3. Results

We screened the titles and abstracts of 800 reports and identified 10 eligible published papers reporting 15 studies [16,17,18,19,20,21,22,23,24] and one review [25]. 

A classification of the different sizing interventions for reducing alcohol consumption and of the identified evidence is shown in Figure 1. Four studies altered serving sizes [22,23,24], eight studies and the review focused on altering glass sizes [16,17,18,25], two studies altered bottle sizes [19,20] and one altered both glass and bottle sizes [21]. Twelve studies and the review focused on wine consumption [16,17,18,19,20,21,23,25] one study focused on beer consumption [24] and two focused on both wine and beer consumption with combined results reported [22]. No other alcoholic drinks were targeted for reduction. All studies, including those that contributed to the review, were conducted in England and by two research groups. Most of the studies were conducted by the group to which the authors of this review are affiliated. 

The characteristics and main results from these 15 studies and one review are summarised in Table 1 and described in more detail below.

### 3.1. Serving Size

#### 3.1.1. Context

The sizes of servings of alcoholic drinks sold in licensed premises in England are subject to regulations [26]. Legal serving sizes of wine by the glass in the UK are 125 mL and 175 mL and multiples of these sizes [26] (Figure 2). A 125 mL glass was once considered the standard size, but this has now been replaced by the 175 mL measure [27]. It is, however, a condition of licenses that 125 mL sizes should be available. The majority of licenced premises also serve 250 mL measures [28]. Draught beer must be legally available to be sold in one of two sizes [26]: pints (568 mL)—which is the most popular measure [29]—and half pints (284 mL). Since 2011, one-third (189 mL) and two-third pints (379 mL) can also be sold, but it is not a legal requirement for licensed premises to make these available [29,30].

Serving sizes of alcohol in licensed premises vary between countries. For example, in French restaurants and bars, the volume of wine poured by the glass is usually between 120 and 150 mL [28], while in the USA, the standard serving size for wine by the glass is 147 mL [31]. For beer, serving sizes range from under 150 mL to 1 litre or more [29]. For example, in the USA, draught beer is usually sold in 118 mL, 236 mL and 473 mL sizes, the latter being the most popular size [32]. In the Netherlands and Belgium, the usual serving size for draught beer is 250 mL, in France, it is 330 mL, and in Germany, 500 mL, depending on the region and type of beer ordered [29]. 

Interventions that focus on altering the serving sizes of alcoholic drinks fall into three categories (Figure 1): i.Those that remove the largest serving size from existing options;ii.Those that reduce the smallest serving size (either by adding a new smaller size or reducing the existing smallest size);iii.Those that add a size smaller than the largest serving size to existing options.

#### 3.1.2. Studies

Three papers reported four studies assessing the impact of altering the range of serving sizes of alcoholic drinks on consumption [22] and sales—a proxy for consumption in settings where the measurement of actual consumption is not feasible [23,24]. Two studies conducted in semi-naturalistic contexts, one in the laboratory and one in a pub room controlled by researchers, removed the largest serving of wine and beer from existing options and replaced them with smaller servings [22]. One study, conducted in licensed premises, reduced the range of existing options by removing the largest serving size of wine by the glass [23]. Another study conducted in licensed premises increased the range of existing options by adding a smaller size for beer that was between the largest and smallest size [24]. 

#### 3.1.3. Findings

##### Wine

Removing the largest serving size is the only intervention type so far evaluated in a naturalistic setting for its potential to reduce wine consumption. This was evaluated in an ABA treatment reversal trial conducted in 21 pubs and bars and restaurants, set over three 4-weekly periods. A was the non-intervention period during which standard serving sizes were available and B was the intervention period during which the largest serving size of wine by the glass was removed (18 premises: 250 mL; 3 premises: 175 mL). The intervention resulted in a reduction of 7.6% (95% CI −12.3%, −2.9%) in the volume of wine sold—a proxy for consumption–with no effect on beer sales or total revenue [23]. 

##### Beer

Adding a serving size smaller than the largest size is the only intervention type so far evaluated in a naturalistic setting for its potential to reduce beer consumption. In an ABA treatment reversal trial in 13 pubs, bars and restaurants, a 2/3 pint option (379 mL) was added for beer and cider sold on tap during the intervention period. This had no impact on the volume of beer or cider sold [24]. 

##### Wine and Beer

Wine and beer consumption in combination has been targeted in two studies that removed the largest serving size and replaced it with a smaller size. In the first of these, conducted in a semi-naturalistic laboratory designed to simulate a home environment, 114 participants were randomised to either large serving sizes of cider (460 mL), lager (460 mL) or wine (165 mL) or to condition in which the large serving size was removed and replaced by a size that had been reduced by 25% (cider/lager: 345 mL; wine: 125 mL). The reduced serving sizes resulted in just over a 20% reduction in alcohol consumption, as measured by recording the number of beverages ordered and by weighing any leftover amounts [22]. 

In the second study, which was conducted in a room in a pub controlled by researchers, 166 participants attended one of four quiz nights, each randomly assigned to either a large serving size condition or a condition in which the largest serving size was removed and replaced by the next available size. The drinks offered were beer, cider and wine, of which the large serving sizes were a pint (568 mL) and 175 mL, respectively. During the reduced serving size nights, the sizes offered were 2/3 pint (379 mL for beer/cider (33% reduction in size) and 125 mL for wine (29% reduction in size). Reducing the services sizes led to a 28% reduction in sales and between and 32% and 40% reduction in alcohol consumption, as measured through direct observation [22].

#### 3.1.4. Summary

Reducing the sizes in which wine and beer are served decreases consumption and sales when the largest serving size is removed and replaced by a smaller size. Adding a serving size slightly smaller than the largest size without removing the largest size does not seem to have an impact. This intervention was used given it did not prove possible to find any pubs, bars or restaurants willing to remove their largest serving size—a pint—from sale for the period of the study. No studies have assessed the impact of adding a size smaller to existing options that is smaller than the existing smallest size.

### 3.2. Glass Size

#### 3.2.1. Context

The size of glasses in which wine is served in licensed premises in England is not regulated, in contrast to the serving sizes in which they are sold by the glass, which is regulated. White wine glasses hold between 236 mL and 355 mL, while red wine glasses hold between 236 mL and >650 mL [33,34]. Wine glass capacity has increased almost seven-fold in England in the last 300 years, from 66 mL in 1700 to 449 mL in 2017. The most marked increase occurred since 1990 [35] (Figure 3). At the same time, the amount of wine consumed in England quadrupled, while the number of wine drinkers remained constant [36]. This suggests that larger wine glasses may account for some of the increase in wine consumption in recent decades.

Beer glasses are designed to hold specific measures when filled to the top. In the UK, the serving sizes in which beer and other ales can be sold are regulated (see section on serving sizes). The size of glassware is therefore regulated and determined by the pre-specified serving sizes and certification is required to ensure they are the appropriate size [37,38,39].

Below we summarise evidence for the impact of glass size on alcohol consumption.

#### 3.2.2. Studies

Five papers reporting nine primary studies [16,17,18,21] and one review [25] assessed the impact of different glass sizes on alcohol sales [16,17,18,25] and consumption [21]. The review [25] combined eight data sets—in a mega-analysis—generated as part of studies reported in three previously published papers [16,17,18]. Eight studies [16,17,18] were conducted in bars and restaurants and one in homes [21]. All targeted wine.

#### 3.2.3. Findings

##### Wine

Between 2015 and 2017, eight studies were conducted in five hospitality establishments in England, aiming to assess the impact of different glass sizes on wine sales. Each study lasted between 14 and 26 weeks and used a multiple treatment reversal design whereby the size of wine glasses (as measured by the volume of liquid they could hold when filled to the brim) was changed fortnightly, while serving sizes of wine—by the glass or bottle—were unchanged [16,17,18]. Capacities of the wine glasses used in the studies were: 250 mL, 290 mL, 300 mL, 350 mL, 370 mL, 450 mL and 510 mL. Data from these studies were combined in a mega-analysis [25], which found that overall sales of wine increased by about 7.3% (95% CI −13.5%, −1.5%) when it was served in larger 370 mL glasses compared with smaller 300 mL glasses. This effect was seen in restaurants but not in bars [25] (Figure 4). 

Just one study to date has assessed the impact of glass size on wine drunk at home. 217 UK households, in which adults were regular wine drinkers, where asked to purchase wine to drink at home. They were given either 290 mL or 350 mL glasses to consume from, as determined by randomisation. On average, 6.5% (95% CI −13.2, 0.3%) less wine was consumed when drinking from smaller than from larger glasses, a difference that was not statistically significant. When taking into consideration whether participants reported any mitigating factors perceived to have affected their wine consumption, such as illness or being away from home, the effect of glass size became slightly larger—6.7% (%95 CI –12.9%, −0.45)—and statistically significant [21].

##### Beer

No studies were found that assessed the impact of the size of glasses on beer consumption.

#### 3.2.4. Summary

Evidence from a mega-analysis of eight data sets generated as part of studies conducted in five pubs, bars, and restaurants suggests that reducing the size of glasses in which wine is served in restaurants decreases wine sales, a proxy for consumption. Using smaller wine glasses at home may also reduce consumption but the evidence for this, which derives from only one study, is less certain. No studies assessed the impact on consumption of serving beer with different sized glasses. 

### 3.3. Bottle Size

#### 3.3.1. Context

Wine is available in a wide range of bottle sizes from 187 mL to the 30 litre Melchizedek [40] (Figure 5). For over 300 years, 750 mL bottles have been the standard size internationally [41]. More recently smaller bottles of 375 mL and 500 mL have become more widely available in many countries, including the UK although these remain in a small minority of wine sold in bottles [42,43,44,45].

Beer is available in a range of sizes which vary between countries. In the UK, a standard bottle of beer contains 500 mL. Smaller bottles usually contain 330 mL. In other countries in Europe, the EU standardised 330 mL bottle is common, although in the Netherlands 300 mL bottles are frequently used. Larger bottles containing 750 mL are popular in Belgium [29]. In the USA, 355 mL bottles are common and larger bottles usually contain 650 mL. In Canada, the standard size is 341 mL [46].

Below, we summarise the available evidence for the impact of bottle size on alcohol consumption.

#### 3.3.2. Studies

Three studies assessed the impact of different bottle sizes on wine consumption [19,20,21], of which one was a small-scale feasibility study [20]. All three were conducted in homes.

#### 3.3.3. Findings

##### Wine

A feasibility study comparing household responses to 750 mL and 375 mL bottles suggested minimal difference in consumption with the two bottle sizes [20]. The results also raised the possibility that 375 mL bottles could, under some circumstances, increase rather than decrease consumption [20]. Having found a source of non-premium wines available in both 500 mL and 750 mL sizes, a study was subsequently set up to assess the impact on consumption of drinking from the standard and a slightly smaller sized bottle.

A cross-over trial was conducted, in which 166 UK households that drank more than two 750 mL bottles a week were asked to buy their usual quantity of wine in 500 mL bottles and 750 mL bottles. The order in which they made the purchases was determined by randomisation. The wine was consumed in homes during each of two 14-day study periods. Households drank about 4.5% (95% CI −7.9%, −1.0%) less wine when using 500 mL bottles, compared with 750 mL bottles [19] (Figure 6). 

An attempt to replicate and extend this study by assessing the impact of wine glass size as well as bottle size was thwarted by the lack of availability of non-premium wines in 500 mL bottles. The supermarket that had stocked the 500 mL bottles for the previous study stopped doing so, probably due to a lack of sales reflecting the non-proportionate pricing of wine in the smaller bottle, i.e., the 500 mL bottle was sold for more than 50/75 the price of the 750 mL bottle. An attempt to replicate the study was therefore carried out with 375 mL bottles, using a cross-over design in which 217 UK households bought their usual quantities of wine in one of the two bottles sizes—375 mL and 750 mL bottles—in an order determined by randomisation. Participants drank wine at home during each of two 14-day study periods. About 3.6% (95% CI −8.3%, 1.1%). less wine was drunk when using 375 mL bottles than when using 75 mL bottles, an effect that was not statistically significant. This may reflect a valid finding of no effect or a study that was underpowered to detect a true effect of bottle size [21]. 

This study also found that when using smaller bottles and smaller glasses together, households consumed on average about 6.5% (95% CI −13.6%, 0.6%) less wine compared to when using larger glasses and larger bottles. Again, there is uncertainty surrounding this combined effect [21].

##### Beer

No studies were found that assessed the impact of the size of glasses on beer consumption.

#### 3.3.4. Summary

Smaller bottles may reduce wine consumption at home but the limited evidence to date suggests this effect is more likely when using 500 mL bottles than 375 mL bottles. The impact of smaller bottles or other containers for drinking beer is unknown.

### 3.4. Potential Mechanisms

The mechanisms by which different sizes of glasses, servings and bottles affect alcohol consumption are largely unstudied. Three possible mechanisms are discussed below.

#### 3.4.1. Affordance

A mechanism that may underlie the effect on alcohol consumption of container size, and specifically the effect of glass size, is affordance, namely what an object or an environment offers an individual [47]. In relation to glasses, it refers to the observation that some glasses, by virtue of some aspect of their design, invite or afford a pattern of behaviour that influences how much is drunk from them [48]. 

When considering glass size, the design feature influencing behaviour is capacity. Larger glasses afford larger pours, by nature of their greater capacity. This then activates what is known as “the portion size effect”, by which the amount we eat or drink depends on the size of the portion or serving we are presented with, leading to increased or decreased consumption, depending on the portion or serving size [9]. Consistent with this, larger wine glasses increase purchasing of wine in restaurants, where more wine is sold by the bottle and therefore free-poured by customers and staff into glasses, but not in bars, where more wine is sold by the glass in fixed serving sizes [25]. Accordingly, one laboratory study found that the larger the glass, the larger the volume of wine that was poured into it [49]. Additionally, supporting this possible mechanism is evidence suggesting that glass size does not appear to have an effect on micro-drinking behaviours—such as sip size—or perceptions of serving size—when fixed volumes of wine are presented in different sized glasses [50]. 

#### 3.4.2. Unit Bias Heuristic

People tend to consume in units of one, regardless of the size of a serving or container: one cup of coffee, one slice of cake, one bottle of wine, one glass of wine and so on [51]. Smaller units may therefore reduce consumption. This “unit bias heuristic” can explain the impact of smaller-sized bottles and servings on consumption, where a bottle or a serving is considered a unit. 

This hypothesis is supported by comments from participants in the bottle size studies conducted in their homes [19,21], who reported that an empty bottle signalled an end of consumption:


*“As its quite normal to just finish a bottle of wine. Whereas it takes more conscious effort to open another bottle. You can just set your limit as a bottle of wine, rather than setting it in terms of cl”*
(Household taking part in the 500 mL vs. 750 mL study [19])

This effect is likely shaped by the effort needed for additional wine: 


*“Having to open a new bottle is a mental hurdle you don’t want to do and it puts you off doing so...”*
(Household taking part in the 500 mL vs. 750 mL study [19])

#### 3.4.3. Size Norms

People hold social and personal norms for what constitutes an appropriate number of portions to consume, as enshrined in the unit bias heuristic described above. They also hold norms for the size of portion to consume [52,53]. These norms are influenced by the portion sizes they routinely encounter day to day. 

In theory, adding an additional smaller size to a range of options could better reflect people’s existing preferences for an ideal size and/or shape [54]. This size norms mechanism could explain why adding 2/3 pints of beer to a range of serving size options had no effect on the volume of beer sold in the study by Mantzari and colleagues [24]. The standard serving for draught beer in the UK is a pint and shifting this could require much longer than the four weeks this was on offer in this study. It could also explain the uncertain evidence for the impact of 375 mL bottles on wine consumption. A bottle containing 750 mL is the most common size in which wine is sold, thereby setting a norm against which smaller and larger sizes are judged. Constituting half the size of a standard wine bottle, as opposed two-thirds which is the case for 500 mL bottles, 375 mL bottles, might just be considered too small [19,21]. This might lead to multiple bottles being consumed per drinking occasion, and/or an increase the frequency of drinking occasions. Indeed, this is supported by comments from some participants in the study assessing the impact of 375 mL bottles [21]:


*“… I did drink on more nights (from smaller bottles) than when I was consuming the larger bottles.”*
(Household taking part in the 375 mL vs. 750 mL study)


*“The smaller bottles were easier to drink and more suited to midweek drinking…”*
(Household taking part in the 375 mL vs. 750 mL study)


*“I preferred the large bottle as I drank less. Drinking one glass from a small bottle seemed like I was leaving hardly any and encouraged me (and my husband) to drink more.”*
(Household taking part in the 375 mL vs. 750 mL study)

#### 3.4.4. Interacting Mechanisms

The above mechanisms likely interact. For example, the affordance mechanism might reduce alcohol consumption in combination with the “unit bias heuristic”, so that people consume a given number of units regardless of the size of the container or serving. Similarly, portion size norms might interact with the “unit bias heuristic” (which is a norm for the number of portions to consume), with people consuming a specific number of units or portions if a container or serving size does not deviate too much from existing size norms. 

The latter could explain any effect of using different sized bottles with different sized glasses. A bottle containing 750 mL is the standard size for a bottle of wine. Given that wine glasses have more than doubled in the last three decades [35], it seems plausible that for most people, their judgements of the standard size have increased correspondingly. When drinking wine at home, people might regulate their consumption both by the number of bottles and glasses they drink. For example, a bottle of wine might be deemed too small because it is deemed to serve too few glasses. This is reflected by comments from a participant in a study when drinking from 500 mL bottles [19]:


*“500 mL… not so great as you only get one and bit glasses each out of it which me and my partner didn’t prefer that as we usually have two glasses of wine each out of one bottle”*


### 3.5. Uncertainties and Future Directions

Reducing the size of servings, glasses and bottles appear to be promising approaches to reducing excess alcohol consumption in populations, to add to more established approaches involving affordability, availability and advertising.

To realise this potential, some uncertainties need reducing to generate evidence that can withstand legal challenge from the industries that stand to lose sales from the implementation of effective alcohol control policies. 

#### 3.5.1. Replicability

Our searches identified only a few studies that assessed the impact of reducing the size of servings, glasses and bottles on alcohol consumption or sales as a proxy for consumption, most of which focused on the size of wine glasses in restaurants. Many of the studies in this review lacked the power to detect other than large effects [16,17,18,20,24,25], and used designs that are at risk of bias [16,17,18,23,24,25], although most attempted to control for possible external variables that can affect drinking patters, such as time of year of the study [16,17,18,19,23,24,25]. The studies need to be replicated in more adequately powered field studies using designs at low risk of bias, in order to elucidate the uncertainties surrounding the effects observed to date. This includes both the presence and the magnitude of the effect of the interventions, especially of using smaller glasses and bottles, singly and in combination, for wine consumption in homes.

It is also unclear whether findings focusing on wine can be generalised to the consumption of other alcoholic drinks such as beer, as there is a complete absence of evidence, highlighting the need for studies to be replicated with other drink types. Similarly, there is a need for findings to be replicated with other container types, apart from bottles, such as cans.

Finally, it is unclear whether findings can be generalised to other countries and populations, as all existing studies were conducted in England and many relied on samples that were predominantly white, of higher education and income and within a narrow age range [19,21] or were conducted in areas with low levels of deprivation [16,17,18,25]. Furthermore, it is unknown how the effects of the interventions reviewed here vary for heavier vs. lighter drinkers. There is therefore a need for the studies reviewed here to be conducted in other parts of the world, with more diverse populations, and designed to assess the effects in those routinely drinking more vs. less alcohol.

#### 3.5.2. Duration of Effects

In most of the studies included in this review the intervention was implemented for two weeks, with the exception of three studies in which it was implemented for four weeks [21,23,24] and two studies in which implemented it for one hour and one evening, respectively [22]. The longer-term effects of reducing the sizes of servings, glasses and bottles on alcohol consumption are therefore unknown. Further studies are needed to assess the effects of smaller sizes of servings, glasses, and bottles beyond the time-period assessed in existing studies, to assess whether any effects are sustained over time.

#### 3.5.3. Compensatory Effects

None of the existing studies assessed the impact of the interventions on total alcohol consumption, i.e., the consumption of all alcoholic drinks. It is therefore unclear whether people compensated for drinking less of the targeted drink –most often wine- by consuming more of other alcoholic drinks. Only two studies attempted to elucidate this by assessing the consumption of a non-targeted drink, i.e., beer and cider when wine was targeted and vice versa [23,24]. These, however, did not measure total alcohol consumption.

#### 3.5.4. Real-World Implications

Although the setting of most existing studies included in this review was naturalistic, the conditions under which some were conducted were controlled, raising questions over their impact in real-world settings.

For example, one of the biggest uncertainties surrounding the use of smaller containers of alcohol such as wine bottles, and smaller serving sizes is their impact on consumption when people are given the choice to select them from a range of options.

In the studies that focused on bottle size included in this review, households were required to adhere to instructions to buy a specific bottle size. It is not known whether they would have selected the smaller bottles had they been given the option to choose a bottle size. Pricing is likely to be an important consideration with proportionate pricing, i.e., half bottles priced at 50% that of full-sized bottles, being key.

Only one study allows for inferences to be made about the real-world impact of increasing the range of options by adding a serving size smaller than the largest. This found no effect [24]. This may reflect some people opting for a slightly smaller size than the largest size they usually select, others opting for a slightly larger size than the one they usually select, or people just completely ignoring the new size.

The above highlights the uncertainty surrounding how best to make smaller sizes effective in reducing consumption in real-world settings. Smaller packages and serving sizes tend to be disproportionately priced compared to larger options and therefore represent less value for money, making it less likely that they will purchase. This is confirmed by participants’ comments in recent studies [19,55]. Additionally, with regard to wine bottles, the availability of non-premium wine in smaller bottles, especially in 500 mL bottles, is limited. This means that there are few ‘like for like’ options for people to make a switch from 750 mL bottles. There is a need for research to assess the impact of increasing the availability and affordability of smaller containers and serving sizes upon selection and total amount of alcohol consumed at a population level.

Although the issue of selection arises when smaller containers or serving sizes are added to existing options, restricting sizes, such as removing the largest serving of wine by the glass or reducing the size of glasses wine is served into, is likely to evoke opposition both from the alcohol industry, given the potential for this intervention to reduce sales of targeted drinks [56], and from the public amongst whom support for such interventions is generally less than for information-based interventions [57]. The impact of industry opposition on policy-makers’ decisions about whether to implement an intervention into policy will in part be modified by the level of public support for the intervention [58,59]. Public support for a policy is, however, amenable to change. For example, communicating the effectiveness of a policy to achieve a valued outcome increases public support [60,61]. Research is needed to explore the acceptability of the various sizing interventions discussed here, as well as methods for increasing possible low levels of support.

### 3.6. Policy Implications

If the effects of reducing the size of servings, glasses and bottles on alcohol consumption are proven reliable with effects sustained over time, these interventions could contribute to existing policies for reducing alcohol consumption across populations. Possible size-related policies include pricing glassware according to capacity, which could increase the demand for smaller glasses for use at home [35]. Capping the size of wine glasses accompanying bottles of wine served in licensed premises also merits consideration. Both these interventions could shift social norms for what constitutes an acceptable size of glass for use in licenced premised as well as at home [35]. Regulating maximum serving sizes for single servings of drinks in licensed premises could again reduce consumption and shift social norms for what comprises an appropriate portion size [52].

Were an effect of smaller bottles on reducing consumption more certain, policies to shift purchasing and consumption to smaller bottle sizes include increasing their affordability. Producing smaller bottles costs proportionately more than the production of larger ones. In order to ensure smaller bottles are proportionately priced in relation to larger bottles, thereby increasing their affordability, fiscal policies would be required that place a higher alcohol tax on larger bottles relative to smaller ones. This would need to be set at rates that discourage consumption, in keeping with the strong evidence that increased affordability of alcohol increases its consumption [62,63].

## 4. Conclusions

Reducing the size of servings, glasses, and bottles are promising approaches to reducing wine consumption across populations. The impact of similar interventions for reducing consumption of other alcoholic drinks, including beer, and of containers other than glasses and bottles, await evaluation. Further studies are also warranted to assess the generalisability of observed effects, including to countries other than England.

## Figures and Tables

**Figure 1 nutrients-14-04244-f001:**
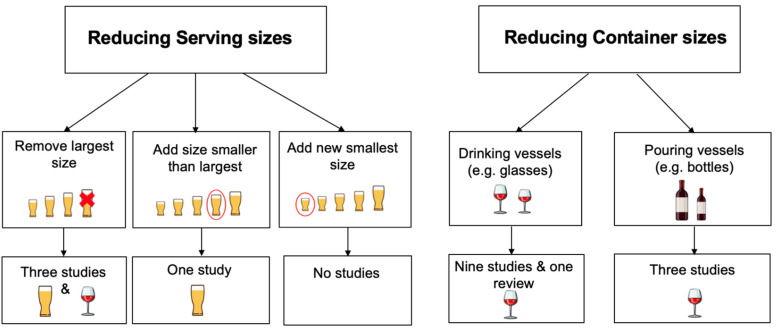
Sizing interventions for reducing alcohol consumption: classifying current evidence.

**Figure 2 nutrients-14-04244-f002:**
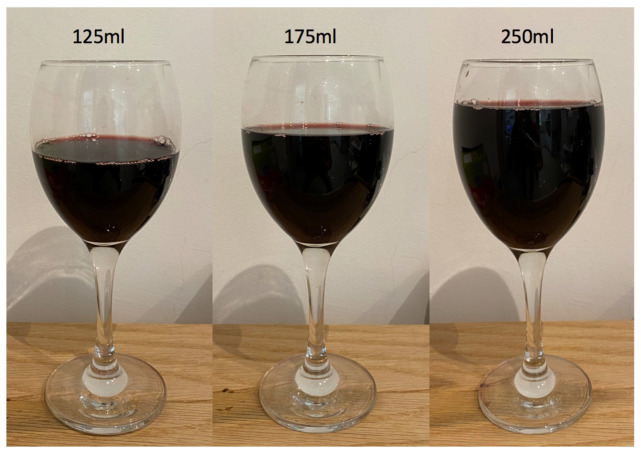
Serving sizes of wine in the UK shown in 355 mL capacity glasses.

**Figure 3 nutrients-14-04244-f003:**
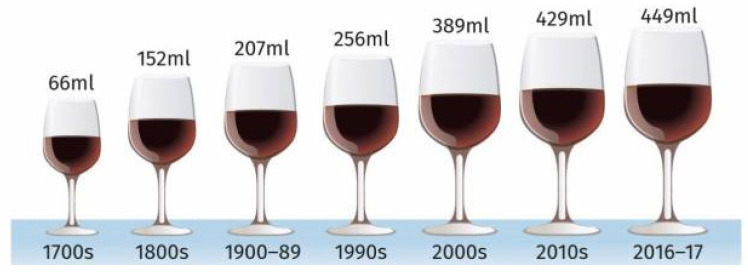
Wine glass capacity since 1700 (source: Institute of Public Health, University of Cambridge).

**Figure 4 nutrients-14-04244-f004:**
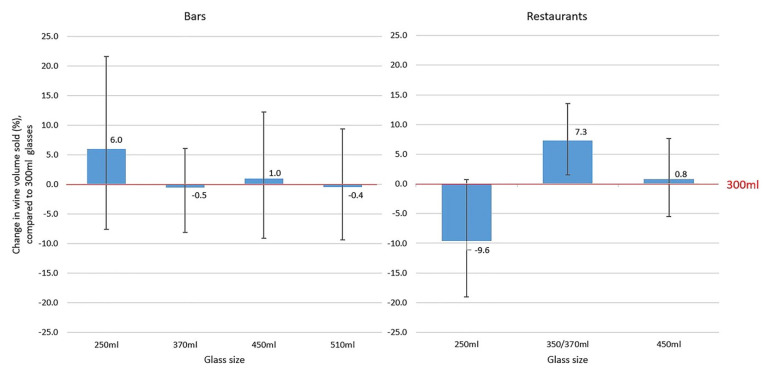
Percentage change (with 95% confidence interval) in daily wine volume sales with each glass size compared to 300 mL glasses in (i) bars and (ii) restaurants (source Pilling et al. 2020 [25]).

**Figure 5 nutrients-14-04244-f005:**
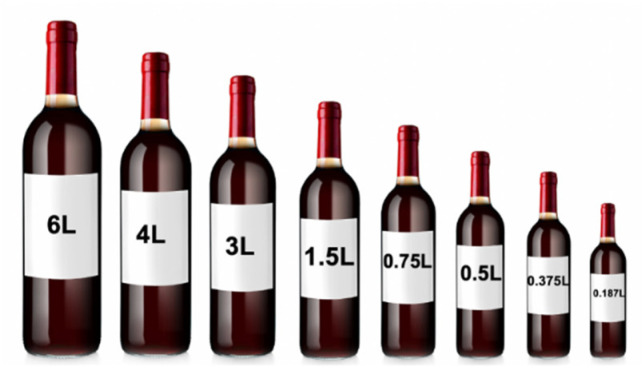
Examples of different bottle sizes of wine.

**Figure 6 nutrients-14-04244-f006:**
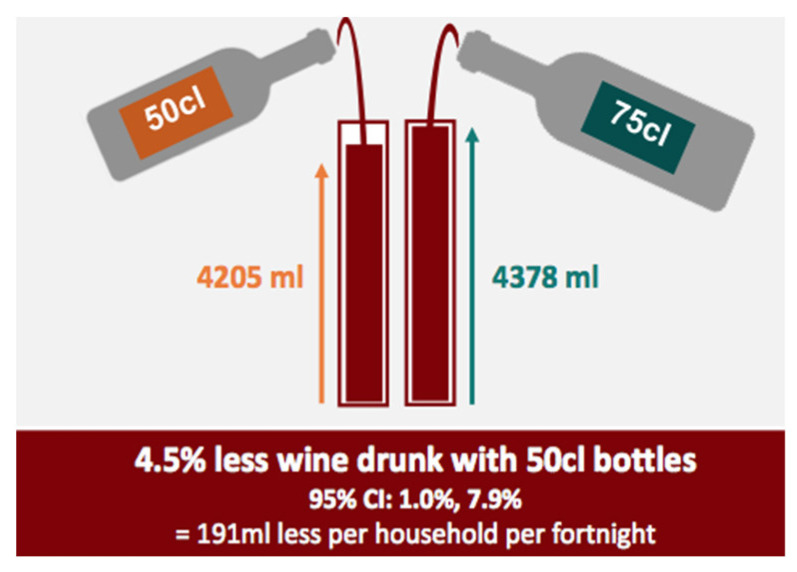
Visual illustration of main results from study by Codling (2020) [19] (Extracted from visual abstract of study available here: https://www.behaviourchangebydesign.iph.cam.ac.uk/wine-bottle-size-and-consumption-in-homes/; access date 26 September 2022).

**Table 1 nutrients-14-04244-t001:** Characteristics of studies included in review.

Serving Size					
Study	Design	Participants/ Setting	Target Drink	Intervention	Main Findings
Mantzari 2022 [23]	Treatment reversal	21 licensed premises	Wine	Removal of largest serving by the glass (most often 250 mL)	Wine sales decreased by −7.6% (12.3, −2.9) with intervention
Mantzari 2022 [24]	Treatment reversal	14 licensed premises	Beer and cider	Adding 2/3 pints to range of options	Beer and cider sales not sig. different with intervention (1.40% [−0.74, 10.4])
Kersbergen 2018 [22]	Experimental	114 participants/laboratory	Beer, cider and wine	Reducing standard sizes by 25%	Alcohol consumption decreased by 20.7–22.3% (95% CI not provided) with intervention
Kersbergen 2018 [22]	Experimental	166 participants/room in pub	Beer, cider and wine	Reducing standard sizes for wine by 29% and for beer/cider 33%	Alcohol consumption decreased by to 32.4–39.6% with intervention (95% CI not provided)
**Glass size**					
**Study**	**Design**	**Participants/** **Setting**	**Target Drink**	**Glass Sizes**	**Main Findings**
Pilling 2020 [25]	Review with mega-analysis (synthesis using raw data)	5 licensed premises	Wine	250 mL 300 mL 370 mL 450 mL 510 mL	Wine sales in restaurants increased by 7.3% (1.5, 13.5) with 370 mL vs. 300 mL glasses. Wine sales not sig different with 250 mL vs. 300 mL glasses (−9.6% [−19.0, 0.7]) or 450 mL vs. 300 mL glasses (0.9%, [−5.5, 7.7]) Wine sales not sig. different by glass size in bars
Clarke 2019 [16]	Multiple treatment reversal	1 licenced premise	Wine	290 mL 350 mL 450 mL	Wine sales increased by 21% (9, 35) with 450 mL vs. 350 mL glasses Wine sales not sig. different with 350 mL vs. 290 mL glasses (−7.4% [−21.6, 9.5])
Clarke 2019 [16]	Multiple treatment reversal	1 licenced premise	Wine	290 mL 350 mL 450 mL	Wine sales not sig. different with 450 mL vs. 350 mL glasses (− 7%, [− 16, 3]) or 350 mL vs. with 290 mL glasses (−7.2% [− 16.5, 2])
Clarke 2019 [16]	Multiple treatment reversal	1 licenced premise	Wine	290 mL 350 mL 450 mL	Wine sales increased by 13% (2, 24) with 350 mL vs. 290 mL glasses Wine sales not sig. different with 450 mL vs. 350 mL glasses (−7.6% [− 17.7, 3.8%])
Clarke 2019 [16]	Multiple treatment reversal	1 licenced premise	Wine	290 mL 350 mL 450 mL	Wine sales not sig. different with 350 mL vs. 290 mL glasses (6%; [−1, 15]) or 450 mL vs. 350 mL (−2.7% [−10.6, 5.9])
Pechey 2017 [18]	Multiple treatment reversal	1 licenced premise	Wine	300 mL 510 mL	Wine sales not sig. different with 510 mL vs. 300 mL glasses (−1.1% [−12.6, 11.9]).
Pechey 2017 [18]	Multiple treatment reversal	1 licenced premise	Wine	300 mL 370 mL 510 mL	Wine sales increased by 10.5% (1.0, 20.9) with 510 mL vs. 370 mL glasses Sales not sig. different with 300 mL vs. 370 mL glasses (6.5% [−5.2, 19.6])
Pechey 2016 [17]	Multiple treatment reversal	1 licenced premise	Wine	250 mL 300 mL 370 mL	Wine sales increased by 14.4 % (3.3, 26.7) with 370 mL vs. 300 mL glasses
Pechey 2016 [17]	Multiple treatment reversal	1 licenced premise	Wine	250 mL 300 mL 370 mL	Wine sales not sig. different with 370 mL vs. 300 mL glasses (8.2 % [−2.5, 20.1])
Mantzari 2022 [21]	Experimental	Home/217 households	Wine	290 mL 350 mL	Consumption reduced by 6.5% (−13.2, 0.3%) with 290 mL vs. 350 mL glasses. Effect not significant
**Bottle Size**					
**Study**	**Design**	**Participants/** **Setting**	**Target Drink**	**Bottle Sizes**	**Main Findings**
Mantzari 2022 [21]	Cross-over RCT	217 households/home	Wine	375 mL 750 mL	Consumption reduced by 3.6% with 375 mL vs. 750 mL bottles (−8.3, 1.1). Effect not significant
Codling 2020 [19]	Cross-over RCT	166 households/home	Wine	500 mL 750 mL	Consumption reduced by 4.5% when drinking from 500 mL vs. 750 mL bottles (−7.9, −1.0)
Mantzari 2020 [20]	Cross-over RCT	16 households/home	Wine	375 mL 750 mL	Consumption reduced by 8.4 mL with 375 mL vs. 750 mL bottles (−596.9, 613.8) Effect not sig. but study was a feasibility study not powered to detect effects

## Data Availability

Not applicable.

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
