# Peer review of "Impact of Sizes of Servings, Glasses and Bottles on Alcohol Consumption: A Narrative Review"

_nutrients, 2022, doi:10.3390/nu14204244_

Round 1
Reviewer 1 Report
This is an interesting summary of limited work on an important topic in a geographical area by basically two research groups. The findings and summaries are not surprising and often barely meet significance so conclusions/ recommendations made with this limited data are hard to accept on face value. The quality of the original work may be in question re. power, interactions with other factors, data collection accuracy, etc and a section detailing these many possible influences/ limitations/ concerns in each study should be presented clearly in the review. Geographical/ ethnic/ local social norms/ time of year/ events and other possible impacts on normal drinking patterns, age/ sex of participants, etc also could provide possible insights on and for the suggested policy changes and a more global assessment of what has been done elsewhere might help shed some light on the possible implications/ suggestions made in the recommendations in this review. It may also be wise to limit recommendations to areas where there is sufficient data (including longer term assessments) and enough studies to ensure these would continue to be accurate in the longer term.
Minor typo/ grammar changes:
Lines 15 and 17 - licensed (not licences?)
Table 1 is very hard to read/ follow as presented
Line 209 - in, England, (extra comma?)
Line 347 - colleauges should be "colleagues"
Line 348 - wording needs clarification - "than the four weeks this was on offer in this." this study?
Author Response
We thank the reviewer for raising these issues. We agree that the quality of the original studies is important and the relevant limitations have perhaps not been adequately highlighted. We have therefore added the following information to the manuscript, under the “Uncertainties” subsection:
Many of the studies in this review lacked the power to detect other than large effects16-18 20 24 25, and used designs that are at risk of bias16-18 23-25 , although most attempted to control for possible external variables that can affect drinking patters, such as time of year the study. 16-19 23-25. The studies need to be replicated in more adequately powered field studies using designs at low risk of bias, in order to elucidate the uncertainties surrounding the effects observed to date. This includes both the presence and the magnitude of the effect of the interventions, especially of using smaller glasses and bottles, singly and in combination, for wine consumption in homes.(page 12)
…all existing studies were conducted in England and many relied on samples that were predominantly white, of higher education and income and within a narrow age range19 21 or were conducted in areas with low levels of deprivation 16-18 25. Furthermore, it is unknown how the effects of the interventions reviewed here vary for heavier vs lighter drinkers. There is therefore a need for the studies reviewed here to be conducted in other parts of the world, with more diverse populations, and designed to assess the effects in those routinely drinking more vs less alcohol. (page 12)
Furthermore, we agree with the reviewer that conclusions should be limited to the areas where there is sufficient evidence. This is why we had already made tentative recommendations only with regards to the impact of serving size and glass size for reducing wine consumption specifically, while also highlighting the need for further research:
Reducing the size of servings, glasses and bottles could reduce wine consumption across populations. The impact of similar interventions for reducing consumption of other alcoholic drinks awaits evaluation. Further studies are also warranted to assess the generalisability of existing evidence (abstract, page 2)
We also specifically mention that effects need to be proven reliable (i.e. replicated in future research) and sustained over time in order for the interventions to have policy implications:
If the effects of reducing the size of servings, glasses and bottles on alcohol consumption are proven reliable with effects sustained over time…(page 13)
Minor typo/ grammar changes:
- Lines 15 and 17 - licensed (not licences?)
Thank you for pointing this out. This has now been corrected throughout the manuscript.
- Table 1 is very hard to read/ follow as presented
This has now been amended in response to relevant suggestions from reviewer 2. We hope that the amendments have improved clarity.
- Line 209 - in, England, (extra comma?)
The comma has been removed
- Line 347 - colleauges should be "colleagues"
This has been corrected.
- Line 348 - wording needs clarification - "than the four weeks this was on offer in this." this study?
The word “study” has been added to the end of the sentence

Reviewer 2 Report
This is a useful and well-written review. The only significant flaw is the design of table 1. The table is difficult to read, it takes up nearly 4 typeset pages, and it has a low density of information. Some suggestions:
· Rename the “Study” column “Ref” and give only the citation number from the references section. Make this the last column.
· Eliminate the “Study Location” column, in which every entry is England.
· Eliminate the “Participants” column.
· Rename the “Serving sizes” column to “Intervention”. Pare down the long entries to show only the major intervention: eg. “Withdraw largest serving size”
· Eliminate the untitled duration column.
· Combine “Outcome measure” with “Main findings”. Give only a brief outcome: eg. “Daily wine sales down 7.6%”.
Follow Journal style guidelines for measured units (https://www.mdpi.com/authors/english-editing) consistently. Put a space between the numeral and the unit symbol: 750 mL (line 250) not 330ml (line 248). Use mL, not ml as the symbol for milliliter. Avoiding cl and cL: 750 mL, not 75 cl.
Some other items the authors may want to think about:
The information provided by ref 35 that the average wine glass in England is 449 mL seems grotesquely implausible. This is more than half a (750 mL) bottle. It is larger than a normal water glass. If 449 ml is average, how big are the 90th percentile glasses?
From the policy perspective, one would need to evaluate the interaction between influence of serving size and the drinker’s usual intake. One might suspect that heavy drinkers will drink until they achieve their customary level of inebriation. Heavy drinkers may also account for much of the alcohol-related morbidity. This is clearly beyond the scope of this review, but the authors may wish to make some mention of additional studies that would be needed to justify a policy change.
Author Response
Reviewer 2
This is a useful and well-written review. The only significant flaw is the design of table 1. The table is difficult to read, it takes up nearly 4 typeset pages, and it has a low density of information. Some suggestions:
- Rename the “Study” column “Ref” and give only the citation number from the references section. Make this the last column.
We appreciate this suggestion but prefer to keep the “Study” column in keeping with the format for such tables suggested by Cochrane Collaboration
- Eliminate the “Study Location” column, in which every entry is England.
This column has now been removed
- Eliminate the “Participants” column.
This has now been removed. Information about the number of participants is now in the “Setting” column
- Rename the “Serving sizes” column to “Intervention”. Pare down the long entries to show only the major intervention: eg. “Withdraw largest serving size”
We have amended the “Serving sizes” column as per the reviewer’s suggestions
- Eliminate the untitled duration column.
This column has now been removed
- Combine “Outcome measure” with “Main findings”. Give only a brief outcome: eg. “Daily wine sales down 7.6%”.
We have combined the “outcome measure” column with the “main findings” column and simplified reporting of the results
- Follow Journal style guidelines for measured units (https://www.mdpi.com/authors/english-editing) consistently. Put a space between the numeral and the unit symbol: 750 mL (line 250) not 330ml (line 248). Use mL, not ml as the symbol for milliliter. Avoiding cl and cL: 750 mL, not 75 cl.
We thank the reviewer for pointing out these guidelines. We have now formatted the manuscript accordingly.
- Some other items the authors may want to think about:
The information provided by ref 35 that the averagewine glass in England is 449 mL seems grotesquely implausible. This is more than half a (750 mL) bottle. It is larger than a normal water glass. If 449 ml is average, how big are the 90th percentile glasses?
The average of 449ml reported in reference 35 was based on measuring the capacity of wine glasses available for purchase in large department stores in England. This included glasses for both white and red wine. The capacity of the latter tends to be quite large. As reported in references 33 and 34 white wine glasses typically hold between 236 mL to 355 mL, while red wine glasses hold between 236 mL to 650 mL +. So, the average of 449ml across glass type is plausible.
- From the policy perspective, one would need to evaluate the interaction between influence of serving size and the drinker’s usual intake. One might suspect that heavy drinkers will drink until they achieve their customary level of inebriation. Heavy drinkers may also account for much of the alcohol-related morbidity. This is clearly beyond the scope of this review, but the authors may wish to make some mention of additional studies that would be needed to justify a policy change.
The reviewer makes an interesting point. The interventions reviewed in the manuscript are those that can be applied at scale, with the aim of influencing behaviour at the population level, regardless of individual characteristics. If inebriation is the outcome of drinking alcohol, one might indeed suspect that heavy drinkers will drink until they have achieved the desired or customary inebriation level, regardless of serving, glass or bottle size i.e. these interventions will not have an effect on these individuals. Most of the studies assessing the impact of serving sizes and glass sizes included in our review were conducted in licensed premises, so it seems plausible that heavy drinkers will have contributed to the data. Although the studies did not assess the moderating role of this individual characteristic, effects were observed across individuals implying that the interventions might be effective when applied at the population level. We have nonetheless added brief mention of the need for studies to assess these interventions amongst heavy drinkers:
Furthermore, it is unknown how the effects of the interventions reviewed here vary for heavier vs lighter drinkers. There is therefore a need for the studies reviewed here to be conducted…designed to assess the effects in those routinely drinking more vs less alcohol(page 13)
